# Dynamics of HIV-1 Gag Processing as Revealed by Fluorescence Lifetime Imaging Microscopy and Single Virus Tracking

**DOI:** 10.3390/v14020340

**Published:** 2022-02-08

**Authors:** Chen Qian, Annica Flemming, Barbara Müller, Don C. Lamb

**Affiliations:** 1Department of Chemistry, Ludwig Maximilians-Universität München, Butenandtstraße 5-13, 81377 München, Germany; chen.qian@lmu.de; 2Center for Nano Science (CeNS), Ludwig Maximilians-Universität München, Butenandtstraße 5-13, 81377 München, Germany; 3Center for Integrative Infectious Diseases Research (CIID), Universitätsklinikum Heidelberg, Im Neuenheimer Feld 344, 69120 Heidelberg, Germany; annica.flemming@gmx.de; 4Nanosystems Initiative München (NIM), Ludwig Maximilians-Universität München, Butenandtstraße 5-13, 81377 München, Germany; 5Center for Integrated Protein Science (CIPSM), Ludwig Maximilians-Universität München, Butenandtstraße 5-13, 81377 München, Germany

**Keywords:** HIV, maturation, gag processing, fluorescence lifetime, single virus tracking

## Abstract

The viral polyprotein Gag plays a central role for HIV-1 assembly, release and maturation. Proteolytic processing of Gag by the viral protease is essential for the structural rearrangements that mark the transition from immature to mature, infectious viruses. The timing and kinetics of Gag processing are not fully understood. Here, fluorescence lifetime imaging microscopy and single virus tracking are used to follow Gag processing in nascent HIV-1 particles in situ. Using a Gag polyprotein labelled internally with eCFP, we show that proteolytic release of the fluorophore from Gag is accompanied by an increase in its fluorescence lifetime. By tracking nascent virus particles in situ and analyzing the intensity and fluorescence lifetime of individual traces, we detect proteolytic cleavage of eCFP from Gag in a subset (6.5%) of viral particles. This suggests that for the majority of VLPs, Gag processing occurs with a delay after particle assembly.

## 1. Introduction

Assembly, release and maturation are critical steps in the late phase of the HIV-1 replication cycle. The viral structural polyprotein Gag plays an important role throughout the entire process. Independently folded domains of Gag are required for membrane targeting (matrix, MA), protein-protein interactions (capsid, CA), packaging of the viral RNA genome (nucleocapsid, NC), and recruitment of ESCRT machinery components that facilitate virion release (p6) [1,2,3]. Gag also recruits the GagProPol polyprotein, which is expressed by ribosomal frameshifting and comprises the viral enzymes PR, reverse transcriptase (RT) and integrase (IN) into the nascent particle. During or shortly after virus release, the mature form of PR generated by autoprocessing from GagProPol mediates proteolytic cleavage of GagProPol and Gag, triggering the morphological rearrangements that yield the characteristic cone-shaped capsid core of the infectious virus [4].

Productive maturation relies on coordination of Gag proteolysis and assembly and on sequential cleavage of specific sites within Gag and GagProPol in an ordered processing cascade [5]. The timing and kinetics of this process are not fully understood. In vitro processing of Gag or virus-like particles using purified PR identified the order of proteolytic events [5,6,7], but does not faithfully recapitulate processing by the endogenous viral PR in situ. Protease inhibitor (PI) wash out of photodestructible PI has been employed to synchronize Gag processing within the context of virus-like particles (VLPs) [8,9,10]. However, this approach uncouples proteolytic maturation from assembly.

Here, we describe the application of fluorescence lifetime imaging microscopy (FLIM) and single virus tracking (SVT) to the analysis of the assembly and maturation of HIV-1. Inserting the fluorophore eCFP as an additional domain into the Gag polyprotein allowed us to monitor changes in its fluorescence lifetime as a readout for proteolytic release. This enabled us to detect Gag processing in relation to the assembly process with high time resolution. We employed confocal laser scanning microscopy (CLSM) with time-correlated single photon counting (TCSPC) to image the assembly and maturation of eCFP-tagged HIV-1 VLPs. Increase of fluorescence intensity was used as a readout to monitor recruitment of Gag.eCFP to nascent assembly sites at the plasma membrane, as described earlier for eGFP [11]. In addition, we exploited sensitivity of the eCFP fluorescence lifetime to the close proximity of other eCFP molecules [12,13,14,15]. By tracking individual assembly sites and VLPs, we show that both incorporation of eCFP into the Gag lattice during assembly and its subsequent cleavage from the lattice during maturation result in measurable changes in the fluorescence lifetime of the fluorophore. Using this method, we were able to observe the timing and kinetics of Gag processing and rearrangement in relation to the assembly process in individually tracked VLPs.

## 2. Materials and Methods

### 2.1. Compounds and Antibodies

Lopinavir was obtained through the AIDS Research and Reference Reagent Program, Division of AIDS, NIAID, National Institutes of Health. Sheep polyclonal anti-CA antiserum was raised against recombinant HIV-1 CA and rabbit polyclonal anti-GFP antiserum was raised against recombinant GFP (both in house).

### 2.2. Plasmids

All HIV-1 plasmids used in this study were based on the non-replication competent derivative pCHIV, which expresses all HIV-1 NL4-3 proteins except for Nef under the control of a CMV promotor and lacks the viral long terminal repeat regions [16]. Plasmids pCHIV^eCFP^ and pCHIV^ieCFP^ are fluorescently tagged derivatives with the eCFP coding sequence inserted between the MA and CA region of the *gag* ORF. pCHIV^eCFP^ has been described previously [17]. pCHIV^ieCFP^ is a derivative of the previously reported pCHIV^iSNAP^ [18]. It was generated by replacing the MluI/XbaI fragment encoding the SNAP-tag by a PCR fragment encoding eCFP flanked by two SQNYPIV protease recognition sites (primer sequences: ggcgc cACGC GTatg gtgag caagg gcgag gagc, cgggc ctcta gactt gtaca gctcg tccat gccga g).

peCFP, eCFP on a pcDNA 3.1 mammalian expression vector was purchased from Invitrogen (Carlsbad, CA, USA). pMA-eCFP was generated by inserting the MA sequence of pCHIV in the NheI/XhoI restriction sites (underlined in the primer sequence) of the peCFP multiple cloning site (MCS) (primer sequences: GATAT AGCTA GCATG GGTGC GAGAG CGTCG G, GGTGC TCGAG TCACT TGTAC AGCTC GTCCA TGCCG).

### 2.3. Cell Lines

HEK293T and HeLa Kyoto cells were cultured in Dulbecco’s modified Eagle’s medium (GlutaMAX, Invitrogen) supplemented with 10% fetal calf serum (FCS; Biochrom), 100 IU/mL penicillin and 100 µg/mL streptomycin at 37 °C and 5%. The HeLa_rCDS_ cell line was generated by Mücksch, et al. [19]. The medium for cultivation of HeLa_rCDS_ cells was supplemented with 2 µg/mL Puromycin and 5 µg/mL Blasticidin. Identity of all cell lines has been authenticated using STR profiling (Promega PowerPlex 21 Kit; carried out by Eurofins Genomics, Ebersberg, Germany). Cells were monitored regularly for mycoplasma contamination using the MycoAlert mycoplasma detection kit (Lonza Rockland, MA, USA).

### 2.4. HEK293T Cell Transfection and Virus Particle Production

HEK293T cells (4 × 10^5^ cells per well) were seeded on a six-well plate 24 h before transfection with the indicated plasmids using a standard polyethyleneimine (PEI) transfection procedure, using 2 µg DNA per well of equimolar mixtures of pCHIV and its labeled derivative (pCHIV^eCFP^ or pCHIV^ieCFP^) and 6 µL 1µg/µL PEI. A final concentration of 2 µM lopinavir (LPV; 10mM) was added to the tissue culture medium at the time of transfection for production of immature particles. At 44 h post transfection (h.p.t.), tissue culture supernatant was collected and cleared via a 0.45 µm nitrocellulose filter. Particles were concentrated from the filtrate by ultracentrifugation through a 20% (*w/w*) sucrose cushion (TL-100 Ultracentrifuge with TLA 45 rotor (Tabletop), Beckman Coulter, Brea, CA, USA, 44.000 rpm for 45 min). Pelleted particles were resuspended in phosphate-buffered saline (PBS). and stored in aliquots at −80 °C.

### 2.5. Quantitative Immunoblot Analysis

Samples were separated by SDS-PAGE (17.5% acrylamide, acrylamide:bisacrylamide 200:1) and transferred to a nitrocellulose membrane by semi-dry blotting. Membranes were blocked with Li-Cor blocking buffer diluted 1:3 in tris-buffered saline with 0.05% (m/V) Tween 20 (TBS-T) and incubated with in-house polyclonal sheep antisera raised against recombinant HIV-1 CA or against recombinant GFP, respectively. Bound antibodies were detected with an Odyssey infrared scanner (Li-Cor, Lincoln, NE, USA), using secondary antibodies, protocols and ImageStudio software provided by the instrument manufacturer. For quantitation of anti-CA reactive bands, purified recombinant HIV-1 CA protein (kindly provided by R. Sahm, University Hospital Heidelberg) was analyzed in parallel.

### 2.6. Virus Particle Preparation for Microscopy

For imaging of free VLPs, 20 µL of HIV-1 particles in PBS were transferred onto the coverslip (Labtek II chambered coverglass, Thermo Scientific, Waltham, MA, USA) and incubated at 23 °C for 15 min. The samples were then washed carefully by removing the storage buffer, adding 20 µL of Dulbecco’s phosphate-buffered saline (DPBS, Gibco) and incubating for 10 min. The wash buffer was then replaced by fresh DPBS for imaging at 23 °C.

### 2.7. Live Cell Sample Preparation

For microscopy experiments, HeLa Kyoto cells (2 × 10^4^ per well) were seeded on 8-well Lab-Tek II chambered coverslides (Thermo Scientific) and incubated at 37 °C for 24 h. Cells were then transfected using Fugene HD DNA transfection reagent (Roche) according to manufacturer’s instructions. An equimolar mixture of pCHIV and its labeled derivative pCHIV^ieCFP^ (in total 70 ng DNA per well) was transfected. The plasmids pMA-eCFP (70 ng DNA per well) or peCFP (20 ng DNA per well) were transfected in a similar manner. Where indicated, a final concentration of 2 µM LPV (kindly provided by Jan Konvalinka) was added at the time of transfection.

To synchronize the start of HIV-1 particle assembly, we used HeLa_rCDS_ cells [19]. HeLa_rCDS_ cells are modified HeLa Kyoto cells in which the concentration of phosphoinositide phosphatidylinositol 4,5-bisphosphate (PI(4,5)P_2_) in the plasma membrane can be controlled via a modified form of the phosphatase 5Ptase. Using a reversible chemical dimerizer, rCD1, 5Ptase is recruited to the plasma membrane where it dephosphorylates PI(4,5)P_2_. The depletion of PI(4,5)P_2_ in the plasma membrane inhibits the assembly of HIV-1 and also leads to the dissociation of pre-assembled Gag lattices at the plasma membrane. By competing out the dimerizer with FK506, cellular hemostasis quickly restores PI(4,5)P_2_ in the plasma membrane and HIV-1 assembly begins almost immediately. HeLa_rCDS_ cells were seeded and transfected as described above with the additional treatment of 1 µM rCD1 (kindly provided by Carsten Schultz, EMBL Heidelberg) 6 h post transfection (hpt) to inhibit assembly. Prior to imaging, culture medium of transfected cells was replaced with imaging medium (DMEM Fluorobrite, Thermo Fisher Scientific) 20–22 h post transfection. Imaging was performed at 37 °C. For HeLa_rCDS_ cells, 1µM FK506 (kindly provided by Carsten Schultz, EMBL) was added to induce virus assembly immediately before image acquisition.

### 2.8. Fluorescence Imaging 

Fluorescence imaging was performed on a home-built confocal laser scanning microscope, as described elsewhere [20]. A NA = 1.49 oil immersion objective (Apo-TIRF 100x Oil/NA 1.49, Nikon, Tokyo, Japan) was used for all measurements. A 440 nm pulsed diode laser (LDH-P-C-440, PicoQuant, Berlin, Germany) was used for excitation at a laser power of between 1–2 µW before objective. Fluorescence emission was detected using a single photon avalanche diode (PDM series, PicoQuant) after a 480/40 nm emission filter. A 30 × 30 µm area divided into 500 lines was scanned over 5 s for each frame. An additional delay of 5 s between each frame was introduced for some cells to reduce photobleaching and phototoxicity over the imaging time.

Raw photon data was processed and analyzed with the software package PIE analysis with MATLAB (PAM) [21]. The pixelwise decay data were transformed using the phasor approach [22,23] (see Appendix A). An aqueous solution of Atto425-COOH was measured at 23 °C and used as a reference to correct for the instrument response function of the system. To simplify representation and analysis, the phase and modulation-derived lifetimes (τ_P_ and τ_M_ respectively) were averaged to produce a single lifetime value.

### 2.9. Single Virus Tracing and Analysis

Raw data was processed by FIJI [24] with a 1-pixel median filter and a 50-pixel rolling ball background subtraction prior to analysis. Particle detection and tracking was performed using Imaris’ Spot detection module (Bitplane AG, Zurich, Switzerland). Within this process, the background was not subtracted and the estimated diameter for spot detection was set to 360 nm. The quality parameter for spot detection was set to 1. Tracking was performed using the Autoregressive Motion algorithm, assuming a maximum distance between frames of 60 nm, allowing for a maximum gap size of 1 and a track duration above 600 s. Filling gaps was disabled. Positions of each detected particle over time were exported to Excel. Exported particle coordinates were subsequently imported into the PAM software to generate image subregions containing each particle, from which the particle-wise phasor values and lifetimes could be extracted.

Mean assembly kinetics were calculated from a selected subset of particle traces. For each trace, the starting point of assembly was manually identified from the intensity increase and fitted with a saturating exponential function. Tracks that can be satisfactorily fitted were selected and synchronized at the starting point of assembly. To obtain the mean assembly kinetics, the mean intensity and lifetime at each frame were calculated and again fitted with the saturating exponential function,
(1)$$y=\left(A_{max}-A_{0}\right)\left[1-\exp\left(-kt\right)\right]+A_{0},$$
where *A_max_* is the maximum fluorescence intensity, *A*_0_ is the initial fluorescence intensity, *k* is the rate of fluorescence increase and *t* is the amount of time that has passed since the start of assembly.

## 3. Results

We first set out to show that the fluorescence lifetime of eCFP can be used to monitor the status of the Gag lattice. For this, we used virus variants carrying an eCFP moiety inserted between the MA and CA domains of Gag. HIV-1 derivatives carrying a fluorophore at this position have been employed previously to track assembly of viral budding sites at the plasma membrane [25]. In HIV^eCFP^ [17], the fluorophore is flanked by a single PR cleavage site (amino acid sequence SQNY↓PIV) at its C-terminus, so that the label will remain attached to the lipid envelope bound MA domain in the mature form of the virus. HIV^ieCFP^ comprises a duplication of this processing site at the N-terminus of eCFP, allowing for release of the free fluorophore upon processing. In both cases, proteolytic processing would result in a change of the environment of the fluorophore from its packed arrangement within the immature shell. Since mature MA remains associated with the viral membrane and was recently observed to retain a modified lattice arrangement [26], the change in the local microenvironment of eCFP is expected to be less pronounced for HIV^eCFP^ compared to HIV^ieCFP^, where the released fluorophore would be able to distribute within the particle volume (Figure 1A). To maintain wild-type assembly properties, eCFP-tagged HIV-1 plasmids were co-transfected with their untagged counterpart (pCHIV) in an equimolar ratio [27]. To determine the effect of virion maturation on the fluorescence lifetime of the particle associated fluorophore, we produced eCFP-labeled viral particles in the presence or absence of the PI lopinavir (LPV). Mature and immature VLPs were purified from the supernatant of transfected HEK293T cells and analyzed by immunoblot with antisera raised against GFP and CA to confirm the inhibition of Gag.eCFP processing by LPV (Appendix A).

Particle preparations were then analyzed by FLIM at room temperature using a home-built CLSM (described elsewhere) [20] (Figure 1B) and fluorescence lifetimes were extracted from the TCSPC photon data using the phasor approach [22,23]. An averaged fluorescence lifetime was calculated for each particle. Two distinct populations were observed based on the eCFP fluorescence lifetime, which we assigned to the mature and immature populations (Figure 1C–E). Immature particles showed comparable eCFP fluorescence lifetimes for both virus derivatives (2.21 (±0.02) ns for HIV^eCFP^ and 2.18 (±0.01) ns for HIV^ieCFP^). Mature HIV^ieCFP^ particles in which the eCFP moiety should be completely released from Gag showed an average increase of 0.45 ns to 2.63 (±0.07) ns compared to immature particles. For the HIV^eCFP^ variant, where the eCFP moiety remains associated with the MA layer, the difference between immature and mature particles was smaller (0.20 ns), with an average lifetime of 2.43 (±0.01) ns for mature particles (Figure 1F). The lifetime of eCFP in the HIV^ieCFP^ particles is still lower than that of free eCFP (3.34 ns, Figure 1C), suggesting that the lifetime of eCFP is sensitive to the high density of protein within the mature virus (based on an average number of 2400 Gag molecules per virion [28], the estimated concentration of eCFP is in the range of ~0.5 mM for an equimolar ratio of Gag and Gag.eCFP).

We also repeated the FLIM experiments on HIV^ieCFP^ VLPs produced in cells transfected with different ratios of pCHIV^ieCFP^ to pCHIV DNA. When the protein density of eCFP within the VLPs was changed by altering the ratio of Gag.ieCFP to unlabeled Gag, we observed a corresponding change the fluorescence lifetime (Appendix A). The mean ratio of labeled to unlabeled Gag protein in VLPs was assessed by immunoblot (Appendix A). Mean eCFP lifetimes of the major VLP population determined for immature HIV^eCFP^ and HIV^ieCFP^, as well as mature HIV^ieCFP^ increased with decreasing proportions of Gag.ieCFP:Gag (Appendix A). Therefore, in addition to the Gag processing state, eCFP lifetime is sensitive to the eCFP molecule density within the particles.

To establish the suitability of HIV^ieCFP^ as a detection system for Gag processing in situ, we characterized the effect of temperature on the fluorescence lifetime of eCFP. We measured a set of HIV^eCFP^ and HIV^ieCFP^ particles at two different temperatures, 23 °C and 37 °C (Appendix A). The fluorescence lifetime of eCFP showed a general decrease when the temperature was increased from 23 °C (Appendix A) to 37 °C (Appendix A), with the effect being greater for mature particles (Δτ ≈ −0.3 ns) than for immature ones (Δτ ≈ −0.2 ns). Nevertheless, the mature and immature populations could still be clearly distinguished for HIV^ieCFP^ particles at 37 °C (Appendix A). We also measured the fluorescence lifetime of eCFP alone and attached to MA in the cytosol of live cells at 37 °C (Appendix A). The lifetime of MA-eCFP (2.51 ± 0.04 ns) was lower than that of eCFP (2.70 ± 0.13 ns), suggesting an influence of the matrix domain on the lifetime of eCFP.

We now focus on the HIV^ieCFP^ variant for live-cell imaging experiments as it provides higher contrast in the lifetime changes between immature and mature virions. FLIM of nascent HIV^ieCFP^ particles was conducted in HeLa Kyoto cells transfected with an equimolar mixture of pCHIV/pCHIV^ieCFP^ (Figure 2A). The coexpression of labeled and non-labeled Gag was confirmed by immunoblot (Appendix A). We first determined the fluorescence lifetime of the particle assemblies at the ventral plasma membrane. The distribution of lifetimes of membrane localized assemblies in cells grown in the absence of LPV was not notably different from the LPV+ condition (Figure 2B,C). In contrast, cell-free particles observed in the vicinity of the producing cells did display the difference in average lifetime as expected for immature vs. mature VLPs (Figure 2D–F). This observation suggests that maturation occurs relatively late in the assembly process such that potential signals corresponding to maturing VLPs at the plasma membrane is obscured by the immature background of nascent or newly assembled VLPs in these still images.

We proceeded to apply single virus tracing (SVT) to follow the assembly of individual VLPs at the plasma membrane (Figure 3A). The combination of SVT with FLIM allowed us to monitor both the fluorescence intensity and lifetime of individual nascent virus buds during assembly of HIV^ieCFP^ in Hela Kyoto cells. From a set of 489 traces with a minimum length of 60 frames (corresponding to 10 min), 354 traces showed an increase in fluorescence intensity over time, as expected for nascent assembly sites [11]. As the number of eCFP incorporated in a VLP increases, we anticipate a decrease in the fluorescence lifetime as observed for immature particles. Indeed, 170 out of the 354 traces displayed a decrease in fluorescence lifetime within the time window of observation.

For analysis, we averaged fluorescence intensities as well as fluorescence lifetimes from these 170 traces upon alignment to the onset of intensity increase and fitted the average intensity and lifetime with a saturating exponential function. The rate of assembly, determined from fitting the plot of the average intensity versus time, was 4.1 (±0.3) × 10^−3^ s^−1^, yielding a half time, τ_1/2_, of 169 s (Figure 3B). The average lifetime decreased with a rate of 7.9 (±1.9) × 10^−3^ s^−1^ (τ_1/2_ = 88 s, Figure 3C), i.e., similar to the rate extracted from the average intensity. The same analysis procedure was applied to 105 traces from HIV^ieCFP^ expressing cells grown in the presence of LPV during transfection and imaging (Figure 3D). Of these, 86 traces showed an intensity increase indicative of assembly. Out of the 86 traces, 51 also showed a concomitant decrease in fluorescence lifetime. The rate of assembly determined by fitting the average intensity of the 51 traces for fluorescence intensity was 6.5 (±2.2) × 10^−3^ s^−1^ (τ_1/2_ = 106 s, Figure 3E); fluorescence lifetime decreased with a similar rate of 4.6 (±3.6) × 10^−3^ s^−1^ (τ_1/2_ = 151 s, Figure 3F). The assembly kinetics as measured here by CLSM are consistent with previously reported values recorded using other microscopy techniques [11,29,30].

The remaining traces, which showed only an increase in intensity but not a concomitant decrease in lifetime, were analyzed in a similar way. The intensity-derived rate of assembly was 5.7 (±0.6) × 10^−3^ s^−1^ for the 184 traces from cells grown in the absence of LPV, and 10.1 (±3.1) × 10^−3^ s^−1^ for the 35 traces from cells grown in the presence of LPV (Appendix A). The mean lifetimes of the traces were 2.1 ns and 2.0 ns respectively. These are comparable to the lifetimes of cell-associated particles (Figure 2A–C), suggesting that the lifetime change was complete before these particles were detected.

Based on our comparison of immature and mature VLPs described above, we proposed that Gag processing and conformation changes in the virion by PR should lead to an increase in fluorescence lifetime in individual particles. Among the 170 traces used for the analysis of assembly kinetics in cells grown in the absence of LPV, 11 traces (6.5%) showed an increase of ~0.4 ns after assembly was complete. This change was similar to the mean shift of lifetime between mature and immature HIV^ieCFP^ virions observed in purified VLPs. An exemplary trace is shown in Figure 4A and in Appendix A (with additional examples shown in Appendix A). In contrast, no such increase in fluorescence lifetime after assembly was observed for traces from cells grown in the presence of LPV (*n* = 51). The same held true when we ignored assembly and looked at the full set of 105 LPV+ traces. (Appendix A). To exclude the possibility that the small number of LPV+ traces caused us to miss potential fluorescence lifetime increase events, we analyzed more cells grown in the presence of LPV. In addition, we applied a newly developed wavelet tracking algorithm [31,32] to generate a large number of traces. Even with the ~1700 traces analyzed using this new approach, we did not observe any traces that showed a lifetime change indicative of Gag in the presence of LPV, while 46 out of 1992 (2.3%) of the traces recorded in the absence of LPV displayed the characteristic increase in fluorescence lifetime (Appendix A).

To estimate the timescale of Gag processing, we analyzed the lifetime signal of the 11 traces identified above where the assembly process was observable. The starting points of the lifetime increase were aligned, and the average lifetime plotted at each time point. On average, the increase in lifetime indicative of Gag processing occurred over a period of ~100 s in an approximately sigmoidal fashion (Figure 4B). We note that the kinetics of Gag cleavage most likely depends on the sequence and surrounding context of the cleavage site.

The mean delay between the start of assembly and maturation as detected by FLIM was 837 (±542) s. However, it is likely that for the majority of VLPs Gag processing is further delayed with respect to viral assembly. The lifetime measurements on VLPs isolated from the tissue culture supernatant (Figure 1E) and recently released VLPs detected in the vicinity of particle producing cells (Figure 2D–F) indicated that the vast majority of VLPs produced eventually underwent Gag processing under the conditions of our experiments. Nevertheless, a distinct shift in lifetime indicative of processing was observed only for a minority of nascent assembly traces during the observation period in the live measurements. Based on this, we can infer that for most VLPs, Gag processing is delayed with respect to viral assembly and occurred beyond the period of observation in our SVT experiments.

## 4. Discussion

We have thus successfully generated and established a system to follow Gag processing in the relation to particle assembly in a live-cell environment. Our results show that the assembly and proteolytic maturation of HIV-1 VLPs induce changes in the fluorescence lifetime of eCFP when it is integrated as part of the Gag lattice. During assembly, the fluorescence lifetime of eCFP decreased concomitantly with the increase in fluorescence intensity, corresponding to an increase in the packing density of eCFP as more and more Gag molecules are recruited to the assembly site. We were also able to detect an increase in lifetime when viral particles transitioned from the immature to the mature state in live-cell SVT experiments. The magnitude of the lifetime change corresponded well with the lifetime difference of 0.4–0.5 ns between immature and mature HIV-1 VLPs measured in vitro.

In our SVT experiments, a small portion of assembly traces (6.5%) showed lifetime changes that indicate maturation, whereas imaging of virions in the vicinity of virus producing cells indicates that the vast majority of virions would eventually mature under our measurement conditions. This is consistent with many EM studies where mature HIV VLPs are observed near the plasma membrane or even in synapses [33,34,35]. The low proportion of maturation associated lifetime changes observed in the SVT experiments suggests that Gag processing primarily occurs after release of the viral particles, which was observed to occur 20–45 min after the onset of assembly in similar experimental setups [11,25]. Due to a combination of several factors such as photobleaching, viral particles moving into regions with dense particle concentrations (Figure 2A) and increased particle mobility following release [11], the tracking algorithm loses the majority of individually tracked particles before or shortly after release. Hence, only the particles that rapidly undergo Gag processing will be detected. The average delay between the onset of assembly and Gag processing for these traces (837 s or 14 min) is biased towards short time scales. However, the low fraction of traces showing maturation suggests that the actual average delay between assembly and maturation is longer.

The increase of lifetime ascribed to proteolytic eCFP release was observed to occur in a relatively short timeframe (~100 s). This is considerably faster than processing kinetics analyzed previously using in vitro systems or purified VLPs [5,8,9,10,36]. However, these measurements employed recombinant Gag or cell free particles, and the readout was completion of the final processing event within Gag (processing at the CA-Sp1 processing site) in the bulk population. In contrast, the proteolytic release of eCFP from Gag monitored here represents a single step in a complex processing cascade and the subsequent redistribution of the marker protein, representing only a part of the overall process.

In summary, we show that the fluorescence lifetime of eCFP can be used in conjunction with SVT to detect viral polyprotein processing in real-time in situ in an asynchronous virus population. As fluorescence imaging and SVT techniques continue to develop, we believe our approach has the potential for further applications in studying the kinetics of viral protease cascades.

## Figures and Tables

**Figure 1 viruses-14-00340-f001:**
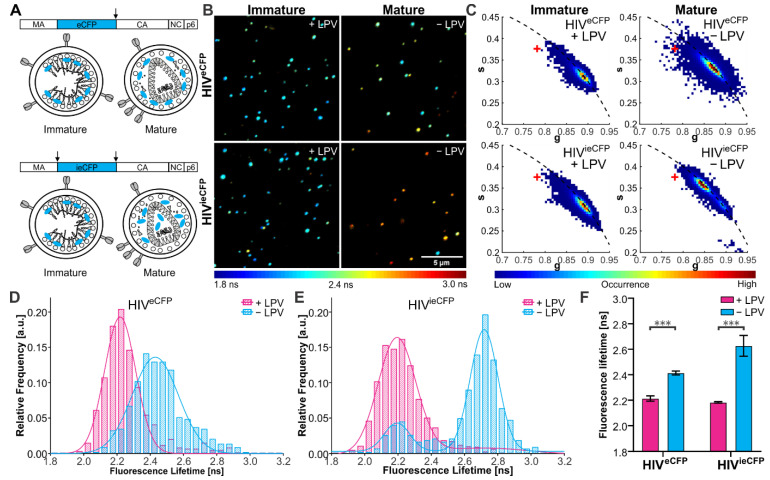
Fluorescence lifetime of eCFP in purified VLPs. (**A**) Scheme of the HIV-1 Gag polyprotein, with eCFP inserted between the MA and CA domains and flanked by one (eCFP, top) or two (ieCFP, bottom) PR cleavage sites. Arrows indicate PR cleavage sites flanking the eCFP domain. Schematic drawing of immature and mature HIV^eCFP^ and HIV^ieCFP^ particles. (**B**) Fluorescence lifetime images and (**C**) phasor plots of immature and mature HIV^eCFP^ or HIV^ieCFP^ particles. A ‘+’ indicates the lifetime of free ECFP measured on the same setup. VLPs were produced in transfected HEK293T cells grown in the presence (immature) or absence (mature) of 2 µM LPV as described in materials and methods and Appendix A. Particles were adhered to borosilicate coverslips and imaged by CLSM with TCSPC. Measurements were conducted at 23 °C. The lifetime determined for individual VLPs is represented according to the indicated color scale. Scale bar 5 µm. (**D**,**E**) Histograms of fluorescence lifetimes of the immature (*cyan*) and mature (*magenta*) particle populations extracted from the phasor analysis. Lines show the fit of the lifetime distributions to the sum of two Gaussians. *n* = 500 particles each. (**F**) Mean and SD of fluorescent lifetime distribution peak from three independent experiments. Statistical analysis was performed using a Welch’s *t*-test (***: *p* < 0.001).

**Figure 2 viruses-14-00340-f002:**
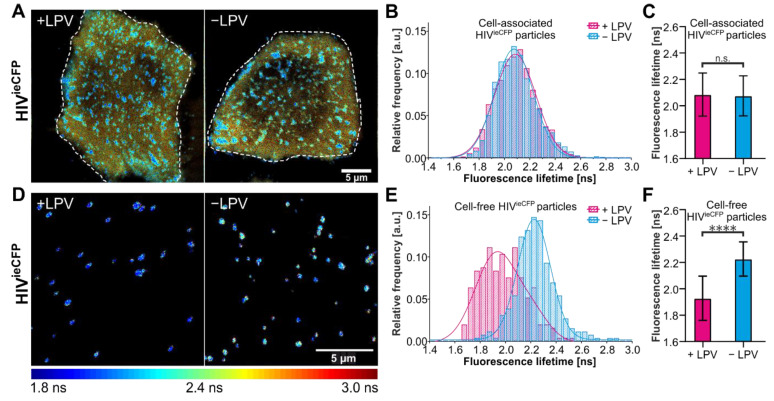
Measurements of the fluorescence lifetime of eCFP in HeLa Kyoto cells. HeLa Kyoto cells were transfected with an equimolar mixture of pCHIV/pCHIV^ieCFP^ and grown in the presence or absence of 2 µM LPV. Cells were imaged 24 hpt at 37 °C by CSLM. (**A**) Fluorescence lifetime images of assembly sites and trapped particles of HIV^ieCFP^ in cells measured in the absence or presence of LPV. (**B**) Histogram of fluorescence lifetime of trapped or cell-associated HIV^ieCFP^ particles and a fit to a single Gaussian distribution (solid lines). *n* = 3196 particles from 7 cells treated with LPV and *n* = 2999 particles from 9 non-treated cells. (**C**) Mean and SD of the fitted Gaussian functions in (**D**). (**D**) Fluorescence lifetime images of released HIV^ieCFP^ particles detected adjacent to transfected cells (cell-free particles). (**E**) Histogram of fluorescence lifetime of cell-free HIV^ieCFP^ particles and a fit to a sum of two Gaussians (solid lines). *n* = 250 particles each from LPV-treated and non-treated cells. (**F**) Mean and SD of the fitted Gaussian functions in (E). The peak of the major species was used. The fluorescence lifetime images are colored according to the ‘jet’ colormap with a range of 1.8 ns–3.0 ns. Scale bars 5 µm. Statistical analysis was performed using a Welch’s *t*-test (n.s.: non-significant, *p* > 0.05; ****: *p* < 0.0001).

**Figure 3 viruses-14-00340-f003:**
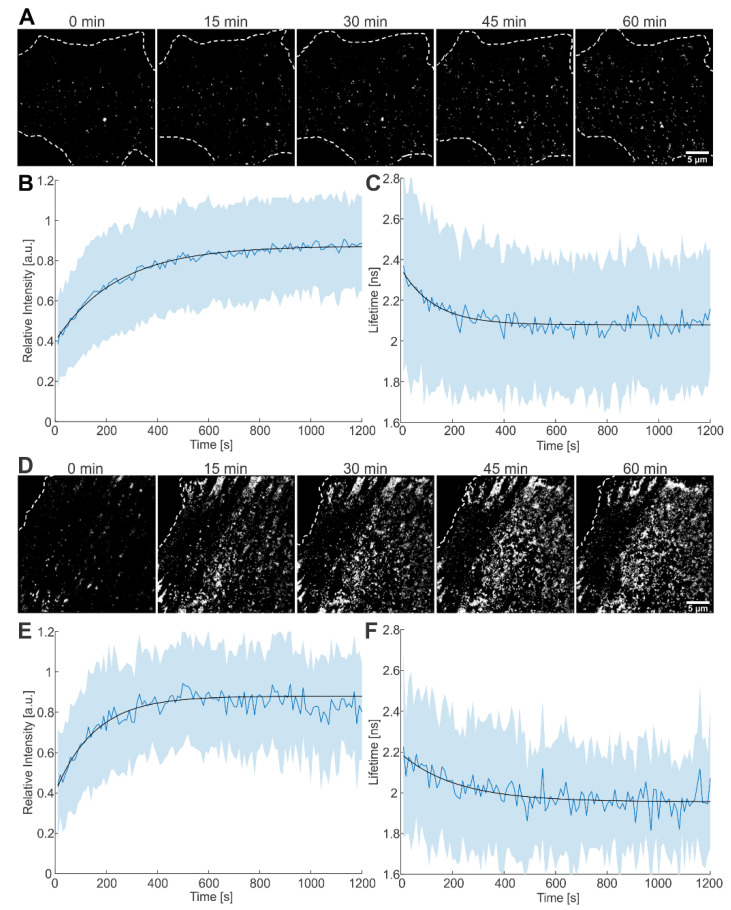
Live-cell imaging of the assembly process at the plasma membrane of HeLa Kyoto cells. (**A**) Time lapse images of Gag-ieCFP assembly recorded at the plasma membrane of a transfected HeLa Kyoto cell grown in the absence of LPV recorded at the indicated times after onset of microscopic observation. Plots of (**B**) fluorescence intensity and (**C**) fluorescence lifetime measured at individual assembly sites as detected in (**A**). Mean values and SD are shown (*n* = 170 sites from 4 cells); black lines represent fits to single exponential equations. (**D**) Time lapse images recorded at the plasma membrane of a transfected HeLa Kyoto cell grown in the presence of 2 µM LPV. Plot of (**E**) fluorescence intensity and (**F**) fluorescence lifetime measured at individual assembly sites as detected in (**D**). Mean values and SD are shown (*n* = 51 sites from 2 cells); black lines represent fits to single exponential equations. HeLa Kyoto cells were co-transfected with equimolar amounts of pCHIV wt and pCHIV^ieCFP^ and imaged at 16 hpt. Cells were imaged every 5 s for 1–2 h. Scale bars: 5 µm.

**Figure 4 viruses-14-00340-f004:**
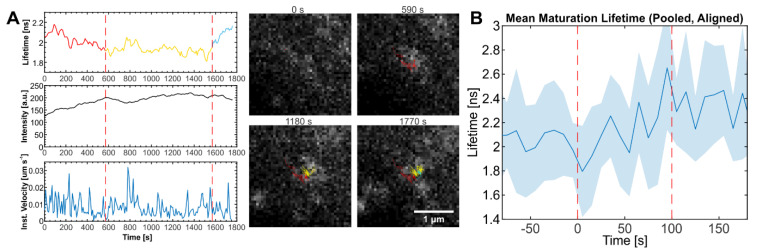
Particles showing eCFP lifetime changes indicative of maturation. (**A**) Lifetime, intensity, and velocity plots of an individual particle showing lifetime changes indicative of maturation following assembly. The lifetime plot is colored by the identifiable state of particle (red: assembly, yellow: plateau phase, blue: maturation). Still images from the movie analyzed for this graph recorded at the indicated times are shown. See Appendix A. (**B**) Mean and SD of fluorescence lifetime of maturing particles in pCHIV/pCHIV^ieCFP^ (1:1) transfected HeLa Kyoto cells. The lifetime of each particle was aligned at the start point of maturation (Time = 0 s) for each trace before averaging. Dotted lines indicate the apparent time range over which the lifetime change occurs. *n* = 11 traces from 4 cells. Scale bar: 1 µm.

## Data Availability

FLIM and SVT data have been deposited in Zenodo (DOI: 10.5281/zenodo.5564327). The PAM software package is available from https://gitlab.com/PAM-PIE/PAM, accessed on 7 January 2022.

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
