# Peer review of "Dynamics of HIV-1 Gag Processing as Revealed by Fluorescence Lifetime Imaging Microscopy and Single Virus Tracking"

_viruses, 2022, doi:10.3390/v14020340_

Round 1
Reviewer 1 Report
The submitted study aims at the studying of the dynamics of HIV-1 Gag Processing revealed by Fluorescence Lifetime Imaging Microscopy and Single Virus Tracking.
The authors used an engineered system inserting the fluorophore eCFP as an additional domain into the Gag polyprotein to allow the monitoring of changes in its fluorescence lifetime as a readout for proteolytic release. In particular, the authors employed confocal laser scanning microscopy (CLSM) with time-correlated single photon counting (TCSPC) to image the assembly and maturation of eCFP-tagged HIV-1 VLPs.
The authors observe an increase of lifetime ascribed to proteolytic eCFP release to occur in a relatively short timeframe, faster than processing kinetics analyzed previously using in vitro systems or purified VLPs. Of note, these measurements employed recombinant Gag or cell free particles, and the readout was completion of the final processing event within Gag in the bulk population. In contrast, the proteolytic release of eCFP from Gag monitored here represents a single step in a complex processing cascade and the subsequent redistribution of the marker protein, representing only a part of the overall process.
In this study the authors show that fluorescence lifetime of eCFP can be coupled with SVT to detect viral polyprotein processing in real-time in situ in an asynchronous viral infection.
Overall this is a technical really nice study with proof of concept techniques. There are also strong supplementary data that support the incorporation of eCFP-labelled Gag in VLP.
Minor comments:
The text on the legend of figures sometimes is hard to read, please increase the characters.
The description of Figure 2A is missed in the text. Figures are not cited in chronological order and sometimes is hard to follow the line of the study.
In the discussion (lines 361,369) I would change HIV-virus particles because it has not been tested a wild type virus.
line 388: change “ in in vitro” “using in vitro…”
Author Response
We thank the reviewer for her/his support of our work.
Minor comments:
The text on the legend of figures sometimes is hard to read, please increase the characters.
As suggested, we have reformatted and increased the font size on all four Figures in the main text.
The description of Figure 2A is missed in the text. Figures are not cited in chronological order and sometimes is hard to follow the line of the study.
We apologize for this oversight. We have now updated all references in the text to correspond to the appropriate figures and panels. We have also moved Supplementary Figure S1B to a new figure, Supplementary Figure S5, so that the figures are now introduced chronologically.
In the discussion (lines 361,369) I would change HIV-virus particles because it has not been tested a wild type virus.
We have changed the text as recommended.
line 388: change “ in in vitro” “using in vitro…”
We have changed the text as recommended.
Reviewer 2 Report
This manuscript describes the use of fluorescent lifetime imaging microscopy, together with single virus tracking, in an analysis of Gag processing in newly released HIV particles. The work is outstanding and a technical tour de force.
My only recommendation for improvement of the manuscript is that it would be helpful to provide a brief explanation, around line 123 and following, of the HeLarCDS system and rCD1. It’s all perfectly clear if one goes to ref. 19, but totally obscure without this. Also, there is a bit of confusion in the labeling of Fig. 2, since the text refers to panels G and H (line 250) but, as the figure is currently labeled, there is no G and H.
Author Response
We thank the reviewer for her/his appreciation of the efforts that went into this publication.
My only recommendation for improvement of the manuscript is that it would be helpful to provide a brief explanation, around line 123 and following, of the HeLarCDS system and rCD1. It’s all perfectly clear if one goes to ref. 19, but totally obscure without this.
We have now provided a brief description of the HeLarCDS system in the Materials and Methods (lines 124-133).
Also, there is a bit of confusion in the labeling of Fig. 2, since the text refers to panels G and H (line 250) but, as the figure is currently labeled, there is no G and H.
We apologize for the confusion. We have now updated all references in the text to correspond to the appropriate figures and panels.